# Tissue Engineering with Stem Cell from Human Exfoliated Deciduous Teeth (SHED) and Collagen Matrix, Regulated by Growth Factor in Regenerating the Dental Pulp

**DOI:** 10.3390/polym14183712

**Published:** 2022-09-06

**Authors:** Vinna K. Sugiaman, Rudy Djuanda, Natallia Pranata, Silvia Naliani, Wayan L. Demolsky

**Affiliations:** 1Department of Oral Biology, Faculty of Dentistry, Maranatha Christian University, Bandung 40164, Indonesia; 2Department of Conservative Dentistry and Endodontic, Faculty of Dentistry, Maranatha Christian University, Bandung 40164, Indonesia; 3Department of Prosthodontics, Faculty of Dentistry, Maranatha Christian University, Bandung 40164, Indonesia; 4Department of Pediatric Dentistry, Faculty of Dentistry, Jenderal Achmad Yani University, Cimahi 40531, Indonesia

**Keywords:** dentin-pulp complex regeneration, signalling molecules, stem cell from human exfoliated deciduous teeth (SHED), tissue engineering

## Abstract

Maintaining dental pulp vitality and preventing tooth loss are two challenges in endodontic treatment. A tooth lacking a viable pulp loses its defense mechanism and regenerative ability, making it more vulnerable to severe damage and eventually necessitating extraction. The tissue engineering approach has drawn attention as an alternative therapy as it can regenerate dentin-pulp complex structures and functions. Stem cells or progenitor cells, extracellular matrix, and signaling molecules are triad components of this approach. Stem cells from human exfoliated deciduous teeth (SHED) are a promising, noninvasive source of stem cells for tissue regeneration. Not only can SHEDs regenerate dentin-pulp tissues (comprised of fibroblasts, odontoblasts, endothelial cells, and nerve cells), but SHEDs also possess immunomodulatory and immunosuppressive properties. The collagen matrix is a material of choice to provide structural and microenvironmental support for SHED-to-dentin pulp tissue differentiation. Growth factors regulate cell proliferation, migration, and differentiation into specific phenotypes via signal-transduction pathways. This review provides current concepts and applications of the tissue engineering approach, especially SHEDs, in endodontic treatment.

## 1. Introduction

Tissue injury can occur when tissue is exposed to various stimuli, including microbial infections, mechanical damage (fractures, cracks, thermal factors), and chemical damage. This condition can cause cell apoptosis or necrosis, as well as microvasculature and stroma damage, leading to the activation of inflammation and wound healing mechanisms. During wound healing, mesenchymal stem cells are recruited to the site of injury to differentiate into stromal cells and replace damaged cells. However, if severe inflammation occurs in the dental pulp, the damaged cells cannot be effectively replaced or healed, a condition called irreversible pulpitis. In this condition, endodontic treatment must be carried out to remove the damaged pulp and prevent the spread of the damage [1,2,3,4].

Endodontic treatment involves partial or complete pulp removal (pulp extirpation) and filling the empty root canal with artificial material. Even so, the endodontic treatment causes the tooth to become more fragile, susceptible to caries and periapical infection and more likely to fracture as the tooth losses its vitality due to the absence of blood supply and innervation [5,6,7,8,9,10,11].

Therefore, it is crucial to maintain the vitality of the pulp. A tooth without a viable pulp loses its defense mechanism and regenerative ability, making it more prone to severe damage and ultimately leading to extraction. Dentin-pulp complex reconstruction is an ideal approach to restoring pulp vitality by using mesenchymal stem cell or progenitor cells and signalling molecules added to the extracellular matrix to recover fibroblasts, odontoblasts, endothelial cells and nerve fiber functions [8,10,11,12,13,14]. Stem cells can be obtained from various tissues, including teeth, buccal mucosa, skin, fat, and bone [15,16]. The pulp of deciduous teeth, rich in stem cells known as stem cells from human exfoliated deciduous teeth (SHED), is a promising, easy-to-get, and noninvasive source of stem cells for tissue regeneration [17,18,19,20,21]. Not only do they have the regenerative ability to generate dentin-pulp tissues but SHEDs also possess immunomodulatory and immunosuppressive properties [20,22].

Scaffolds are 3-dimensional microstructural materials that provide a biological environment and structural support to facilitate cell growth, desirable interactions, and the formation of functional tissues [8,23,24]. One popular scaffold material is collagen. Collagen is a natural extracellular matrix built from protein and abundant in hard and soft tissues [23]. Collagen is biocompatible, permeable, and biodegradable, so it can function in helping migration, adhesion, proliferation, and cell differentiation [8,12].

Growth factors are polypeptides that play a very important role in the signaling process that occurs during tissue formation and regeneration of the dentin-pulp complex [25,26]. In the dentin-pulp complex regeneration, several growth factors work together through different signalling mechanisms, including Transforming Growth Factor-β (TGFβ), Vascular Endothelial Growth Factor (VEGF),Bone Morphogenic Protein (BMP), Fibroblast Growth Factor (FGF),Platelet-Derived Growth Factor (PDGF), and Nerve Growth Factor(NGF) [25,27,28]. Growth factors will bind to cell surface receptors that subsequently induce cellular processes such as cell proliferation, angiogenesis, neovascularization, and all important steps in the regeneration process [28,29].

Growth Factorplays a role in various stages of the healing process and tissue regeneration, including cell migration, angiogenesis, and neurogenesis [26]. It can also induce odontogenic differentiation through ALK5/Smad2/3, TAK1, p38, and MEK/ERK signalling pathways, supporting cell proliferation and collagen formation [30,31].Tissue engineering applications in endodontic treatment are expected to replace damaged or lost tissue with new natural pulp tissue and reduce the use of artificial materials, making teeth fully functional again [14].

## 2. Tissue Engineering (TE) in Endodontic Treatment

As mentioned before, one challenge in endodontic treatment is maintaining dental pulp vitality and preventing tooth loss. Regenerative endodontics can overcome this hurdle [32]. According to the American Association of Endodontists, regenerative endodontics is a procedure designed based on biological principles to physiologically replace damaged tooth structures, including root and dentin structures, as well as cells in the pulp-dentin complex [10,32,33,34].

There are two concepts in regenerative endodontics, namely [35]:(1) guided tissue regeneration (GTR), also known as the revascularization or revitalization approach, and (2) tissue engineering (TE), an interdisciplinary approach to repairing damaged tissue using by combining three components: (1) cells (especially stem cells) capable of forming pulp tissue, root dentin, and tooth-supporting tissues, (2) scaffolds to facilitate cell proliferation and differentiation, and (3) bioactive molecules (generally growth factors) as shown in Figure 1 [28,35,36,37,38].

## 3. Stem Cells

Stem cells are unique cells that possess self-renewal and differentiation properties into another cell type. Based on their differentiation potency, stem cells are divided into the following groups [39,40,41,42].

### 3.1. Totipotent Stem Cells

Totipotent stem cells are stem cells that can generate all types of cells and tissues that exist in organisms and can usually be obtained from embryonic stem cells (from embryos 1–3 days old). Totipotent cells have the highest differentiation potential and allow cells to form embryonic and extra-embryonic structures. An example of a totipotent cell is the zygote, formed after a sperm fertilizes an egg. These cells can later develop into one of the three germ layers or form the placenta. After about four days, the cell mass in the blastocyst becomes pluripotent. This structure is a source of pluripotent cells [35,43].

### 3.2. Pluripotent Stem Cells

Pluripotent stem cells are stem cells that can generate most cell types (over 200) and tissues found in organisms and have the ability to differentiate into cells of ectodermal, mesodermal, and endodermal origin. They can be obtained from a 5–14 day old blastocyst [35,44,45].

### 3.3. Multipotent Stem Cells

Multipotent stem cells are stem cells that can generate a limited number of cell and tissue types depending on their origin. These cells can be obtained from cord blood, fetal tissue and postnatal stem cells, including dental pulp stem cells [35,45,46].

### 3.4. Unipotent Stem Cell

Unipotent stem cells are stem cells that have the narrowest differentiation ability; the can only differentiate into one cell type but are able to divide repeatedly [43,45].

### 3.5. Induced Pluripotent Cells

Induced Pluripotent Cells are pluripotent stem cells formed by the induction of multipotent cells or adult somatic cells with pluripotent factors such as Oct4, Nanog, Sox2, Klf4, and C-myc [45,47].

There are two approaches to delivering stem cells into the root canal. The first approach is cell transplantation, where autologous or allologous stem cells are applied directly to the root canal. The major obstacle to this process is the immune rejection of allologous stem cells. The second obstacle is cell homing, where stem cells are sent to the injured area; this process is influenced by many factors, such as age, cell number, culture conditions, and method of application. This condition involves the use of chemotactic factors such as stromal cell-derived factor (SDF)-1 are injected into the site of injury to induce stem cell migration from the periapical area to the root canal [27,48].

Based on their stage of development and origin, stem cells can be broadly classified into [32,35,41,47]: (1) embryonic stem cells, which are stem cells derived from embryos, mainly from blastocysts. These cells are capable of dividing and renewing themselves over a long period; (2) adult stem cells, which are stem cells derived from postnatal tissue, can be isolated from various body tissues, such as bone marrow, adipose tissue, encephalon, epithelium, dental pulp, etc.

Tissue injury is always associated with the activation of the immune system or inflammatory cells, including macrophages, neutrophils, CD4+ T cells, CD8+ T cells, and B cells, triggered by cell apoptosis, necrotic cells, microvascular damage, and stroma [40,49,50,51]. Mesenchymal stem cells can regulate specific and non-specific immune systems by suppressing T cells and dendritic cell maturation, decreasing B cell proliferation and activation, inhibiting NK cell proliferation and cytotoxicity, and increasing T regulatory (Treg) cell formation [49,50].

There are two mechanisms of stem cell immunomodulation: soluble factor secretion and cell-to-cell direct contact. Prostaglandin E2 (PGE2), indoleamine 2,3-dioxygenase (IDO), nitric oxide (NO), interleukin-10 (IL-10), hepatocyte growth factor (HGF), and transforming growth factor 1 (TGFβ1) are secreted factors that have immunomodulatory properties. The cell-to-cell direct contact mechanism involves CD274 (programmed dead ligand 1), vascular cell adhesion molecule-1, and galectin-1 expression. These molecules reduce effector T cell proliferation and increase the proportion of regulatory T cells (Treg) [49,50,52].

Various stem cells can be found in teeth and their associate tissues, such as stem cells from human exfoliated deciduous teeth (SHED), dental pulp stem cells (DPSC), stem cells from the apical papilla (SCAP), periodontal ligament stem cells (PDLC), dental follicle precursor cells (DFPC), dental papilla cells (DPC), dental mesenchymal stem cells (DMSCs), and dental epithelial stem cells (DESCs). For pulp regeneration purposes, SHED, DPSC, and SCAP have strong potential [35,41,53,54,55].

## 4. Stem Cells from Human Exfoliated Deciduous Teeth (SHED)

Stem cells from human exfoliated deciduous teeth (SHED) were first obtained by Miura et al. in 2003. SHED expresses cell surface markers STRO-1, CD10, CD29, CD 31, CD44, CD73, CD90, CD105, CD146, CD13, CD166, Nestin, DCX, -tubulin, NeuN, GFAP, S-100, A2B5, CNPaseNanog, Oct3/4 and SSEAs (-3, -4) and does not express CD14, CD15, CD19, CD34, CD45, and CD43 [41,56,57,58,59].

SHEDs have two major advantages compared to other stem cells derived from dental tissue: they are easier to gain through noninvasive procedures and have a high proliferation rate [34,41,56,60,61]. SHEDs exhibit higher proliferation rates compared to dental pulp stem cells (DPSCs) and bone marrow-derived mesenchymal stem cells (BMMSCs) [41,45,58,62,63,64].

SHEDs possess higher potential in forming dentin-pulp complex cells, namely osteoblasts, chondroblasts, adipocytes, endothelial cells, nerve cells, and odontoblasts [57,58,65,66,67].The ability of SHEDs to differentiate into odontoblasts is characterized by the expression of dentin matrix protein-1 (DMP-1) and dentin sialophosphoprotein (DSPP) [45,58]. DSPP induces stem cells to odontoblast differentiation through SMAD 1/5/8 phosphorylation and nuclear translocation via the P38 and ERK1/2 pathways. DMP-1 involves maintaining dentin mineralization [68,69].

As for the potential for neural regeneration, SHEDs show more intensive expression of neural differentiation markers than DPSCs, such as b-III-tubulin, and nestin, in neural induction cell culture [37]. SHEDs are also able to increase the angiogenesis process by forming vascular connective tissue structures and expressing and synthesizing VEGF [70].This ability is crucial to maintaining pulp viability as it can supply oxygen and nutrients needed for cell metabolism for tissue regeneration [71].

SHED also functioned as an immunomodulator by suppressing T helper 17 (Th17) cell function and upregulating CD206+ M2 macrophages [57,62]. SHEDs are able to induce the secretion of proinflammatory cytokines, such as interleukin 1b (IL-1b), interleukin 6 (IL-6), interleukin 10 (IL-10), and tumor necrosis factor- a. SHEDs are also capable of inhibiting lymphocyte CD178 expression, suppressing the proliferation of lymphocytes, and decreasing the secretion of IL-4 and IFN-g while sequentially increasing the number of T-reg cells [37,72,73].

## 5. Collagen Scaffold

Scaffolds are required for regeneration or tissue engineering to facilitate cell growth and functions in the transplanted area [74,75,76]. Interaction of the cell with the extracellular matrix influences many signalling pathways that change cell behaviours, i.e., adhesion, proliferation, and differentiation [76,77]. Scaffolds can be made of both natural and synthetic materials. Nanoscale proteins are the primary natural scaffolding materials. Nanoscale proteins include collagen, fibronectin, and vitronectin. Synthetic polymers are popular materials because they are biocompatible, biodegradable, mechanically stable, and can be designed in a variety of compositions and shapes [77,78]. These properties enable polymers to biologically affiliate and mimic the natural cell-extracellular matrix [76,79]. Natural scaffolds, such as collagen, have better biocompatibility, whereas synthetic polymers can be controlled for their physicochemical properties, such as their solubility, microstructure, and mechanical strength [76,79].

Nanofibrous scaffolds are more popular than microfiber scaffolds due to their high surface area, interconnected porosity, and positively stimulating extracellular cell-matrix interactions [76]. Nanofibrous scaffolds are made by three methods, namely electrospinning, self-assembly, and separation phase [77]. Electrospinning is the tissue engineering application method most frequently used to synthesize collagen or synthetic scaffolds and/or transport systems for drugs [76].

Collagen is a hydrogel material with high biocompatibility, viscoelasticy similar to soft connective tissue, the ability to transport nutrients and waste, uniform cell encapsulation, in situ gelation ability, and compatibility to be modified by biofunctional molecules or growth factors [80]. Collagen contains arginine-glycine-aspartic acid (RGD) adhesion ligands, which enable cell-biomaterial interactions, leading to cell adhesion [75]. Collagen matrices are compatible with dental pulp stem cell proliferation, adhesion, and differentiation, as shown by the formation of capillary-like microvessels [76,81,82]. Two commercial injectable scaffolds, self-assembling peptide hydrogel and rHCollagen type I, were evaluated. It was found that both of those scaffolds promote SHED cell survival, and when injected into the root canal, these materials promoted odontoblast putative marker expression [83].

Different collagen materials have been compared, such as collagen type I and III, alginate, and chitosan, generating a good result in the proliferative and mineralizing activity of type I collagen. After implanting these cells, the formation of vascularized pulp-like tissue, odontoblast-like cells, and new dentin is produced. SHEDs adhere to PLA cells in dentinal discs [80].

Collagen is a biocompatible material that can be degraded by enzymes; however, natural polymers are difficult to produce and may transmit pathogens from animals (as they are usually produced from animal products) or stimulate an immune response. No scaffold materials have ideal structures and properties that totally resemble natural extracellular matrix as natural ECM comprises complex architecture made up of structural proteins (collagen and elastin), specialized proteins, and glycosaminoglycans. This architecture provides not only structural support for tissue but also a selective dynamic environment that is remodeled via biochemical signals to direct cellular responses [84]. A scaffold should combine the best properties of biomaterials and be as close to the physiological environment of the ECM as possible [80].

## 6. Growth Factor as Regulator

Regulating molecules are required for SHED to generate endothelial cells, odontoblasts, and neurons that will form the dentin-pulp complex architecture [71,85,86]. They work in signal transduction pathways to regulate cell proliferation, migration, and differentiation into specific phenotypes. BMPs, PDGF, FGF, TGF, EGF, and IGFs are the most common WNT proteins [87,88,89].

VEGF stimulates SHEDs to undergo endothelial cell differentiation. In an experiment described by Annibali (2014), SHED was incubated in an endothelial cell growth medium (EGM-2MV). This medium contains ascorbic acid, hydrocortisone, rhEGF, FBS, R3-IGF-1, rhbFGF, rhVEGF, and VEGF [71,85]. MEK1/VEGF/ErK, Wnt/VEGF/-catenin, and Notch-EphrinB2/VEGF-DLL4 signaling pathway regulation in response to VEGF stimulation and the expression of VE-Cadherin (endothelial markers), VEGFR2, and CD31 increased dramatically [71,85]. Furthermore, the endothelial-like cells generated by SHEDs could anastomose with the host vascular network, which was demonstrated by an experiment using LacZ tags and galactosidase staining [85].

Odontoblast differentiation was observed after BMP-2 stimulation. This regulatory molecule involves the production of tubular dentin, odontogenesis and morphogenesis. Dentin sialophosphoprotein (DSPP) marker will be abundantly expressed for this distinction [85,90,91,92]. The production of DSPP is also influenced by two catalytic subunit signaling complexes that target rapamycin complexes 1 and 2 (TORC1 and TORC2). TORC1, which is also required for protein synthesis and translation, regulates and directs cell cycle, growth, and proliferation. Suppression of TORC1 prevented mineralized matrix deposition, which also severely limited the synthesis of DSPP. TORC2 influences both cell survival and cytoskeleton rearrangement. Inhibition of TORC2 promoted mineralization [85,93].

SHED culture in DMEM supplemented with vitamin D3, ascorbic 2-phosphate, dexamethasone, and glycerol phosphate resulted in the expression of odontoblast-specific genes, DMP1 and DSPP. Culture also showed mineralized matrix as visualized using Alizarin red [85,94].

Different techniques for isolating SHEDs revealed various traits for odontoblast differentiation. Despite having functioning odontoblast phenotypes, SHEDs isolated by direct outgrowth showed a decreased rate of mineralization and abnormal cell elongation and polarization due to the vertical orientation of the cell body alongside the dentin-like matrix. SHEDs isolated using enzymatic dissociation quickly formed mineralized tissue and kept their spindle-shaped morphology [85,90]

In immunocompromised mice, the ability of SHEDs to develop into odontoblasts was examined. The dorsum of subcutaneous tissue was implanted with ceramic tricalcium phosphate/hydroxyapatite (TCP/HA) powder and SHED combinations [85].

This resulted in the formation of dentin-like structures. However, the transplant could not form a complete dentin-pulp-like complex. Only 25% of the clones from one of the colony-derived SHED strains transplanted were found to produce ectopic dentin [85].

In another study, slices of extracted third molar teeth were used. To create a porous biodegradable scaffold, poly-L-lactic acid was used to fill the pulp chamber, which was in close contact with the predentin layer. After 1428 days, cells adjacent to the predentin exhibited an active dentin-secreting odontoblast. DSP was also expressed. The cell nuclear location is thought to be polarized eccentrically. The cells displayed cell-cell gap junctions, a well-developed rough endoplasmic reticulum, the Golgi complex, and a large number of vesicles [85].

SHEDs have also been confirmed to be able to develop into neurons. Several neuronal markers, including glutamic acid decarboxylase (GAD), III-tubulin, nestin, 2′,3′-cyclic nucleotide-3′phosphodiesterase (CNPase), tyrosine-hydroxylase (TH), polysialylated-neural cell adhesion molecule (PSA-NCAM), and glial fibrillary acidic protein (GFAP) were expressed by SHED-derived neurons.10–12 Several cytokines, including FGF8, SHH, bFGF, and GDNF, influence SHED neuronal regeneration [86,95,96].

FGF8 is responsible for the dorsalization of the anterior neural tube [96]. The notochord secretes SHH during development to induce a general ventral cell destiny in order to generate floor plate and motor neurons. bFGF acts as a proliferation and differentiation regulator. After five days of culture on poly-L-lysine coated dishes without serum, the cells rapidly lost their mesenchymal appearance and took on a more neuronal appearance, including neurite-like outgrowth. Continued injection of SHH/FGF8 generated neurons with developed and extended axon- or dendrite-like structures [85,96].

Upregulation of lncRNA C21orf121 and the downregulation of miR140-5p aid in the differentiation of SHEDs into neuronal cells. lncRNA C21orf121 prevents BMP2 from binding to miR140-5p, which subsequently increases BMP2 production and promotes SHED neurogenesis [86,97]. Table 1 shown several researches that have been conducted using tissue engineering technology in pulp regeneration.

## 7. Dentin Pulp Regeneration

Dentin pulp regeneration aims to revitalize necrotic, infected, or lost pulp teeth by restoring the morphology and function of the pulp. Ideal pulp regeneration should possess natural structures such as nerve fibers and blood vessels, allowing nutritional, defense, sensation, and immunological functions to be restored [10,111]. Growth factors, scaffolds, plasma, or other associated cells such as dentin/odontoblasts, fibroblasts, or endothelial cells may provide regenerative signals in this regeneration process, resulting in cell migration, proliferation, differentiation, angiogenesis and extracellular matrix deposition [28,112].

Endothelial cells differentiate into mesodermal precursor cells (angioblasts) during vasculogenesis, whereas new blood vessels are formed from previously existing blood vessels during angiogenesis. VEGF is the main regulator of angiogenesis and can also increase vascular permeability [28,113]. FGF, another growth factor with an angiogenic role, can attract DPSCs to migrate and proliferate [28]. PDGF can significantly boost cell proliferation, angiogenesis, and odontoblast differentiation [114,115]. BMP7 promotes the formation of dentin (dentinogenesis) [116].

Nerve growth factor (NGF) plays an important role in the nervous system’s growth, differentiation, and defense mechanisms by preventing apoptosis and reducing neuronal degradation. NGF expression is typically increased in damaged and developing teeth; this growth factor promotes the proliferation of sensory and sympathetic nerve cells [28]. NGF is also involved in the processes of angiogenesis by inducing VEGF upregulation. NGF binds to tyrosine kinase receptor (TrkA) on the cell surface, resulting in TrkA phosphorylation and activation of multiple signaling pathways, including PI3K/Akt, Ras/Raf/MEK/ERK 1/2, and PLC/PKC. Activation of each of these pathways results in a variety of biological functions, including the prevention of apoptosis [117,118,119].

In this review, we focus on regenerative endodontic treatment using SHED, collagen scaffold, and growth factors to regenerate dental pulp tissue through tissue engineering technology. The concept of tissue engineering is expected to answer the challenges in dentistry in maintaining the vitality of the dental pulp. Various studies and research are being continuously carried out in order to obtain the best strategy in tissue engineering and regenerative endodontics. This is achieved by understanding the behavior of cells, the suitability of the material with the scaffolds, as well as the growth supporting factors for each specific tissue or organ to be created; these factors are the keys to the success of tissue engineering.

## 8. Conclusions

In responding to the challenges in dentistry to maintain pulp tissue and prevent tooth loss with irreversible or necrotic pulpitis, regenerative endodontics utilizing tissue engineering technology can be developed. In this technology, the utilization of SHEDs, which have excellent potential with high proliferation speed and ability to differentiate into various cell-forming dental pulp cells, collagen scaffolds as a medium for cell growth and function, and growth factor as a regulator can be utilized to repair and regenerate pulp tissue by regenerating pulp tissue naturally to be fully functional again.

## Figures and Tables

**Figure 1 polymers-14-03712-f001:**
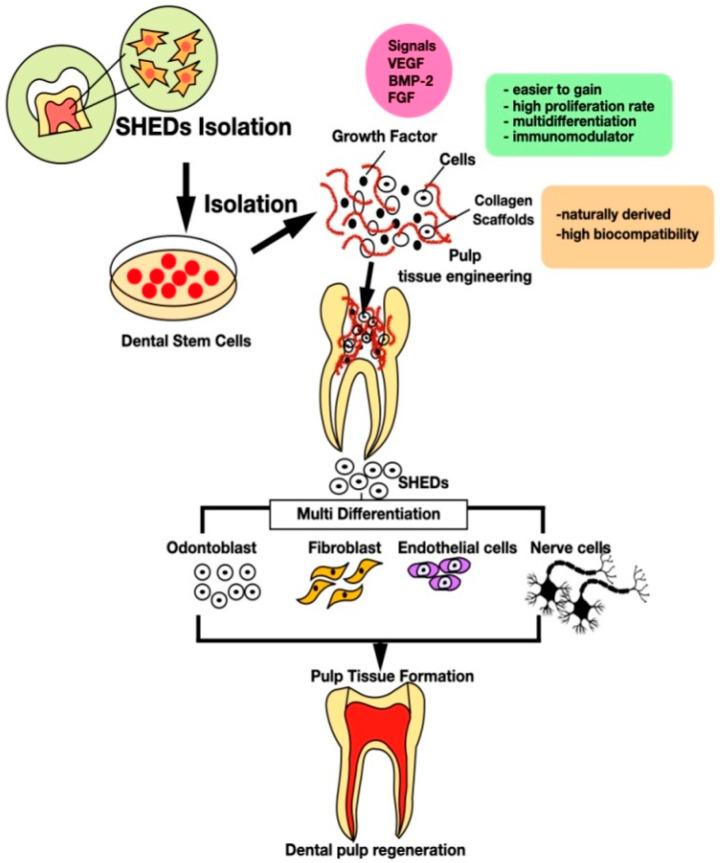
Tissue engineering technology in dental pulp regeneration.

**Table 1 polymers-14-03712-t001:** Stem cells for dental pulp regeneration [83,98,99,100,101,102,103,104,105,106,107,108,109,110].

Article (Author, Year)	Type of Stem Cell	Type of Scaffold	Types of Studies	Evaluation Technique	Outcome
Cordeiro, 2008 [98]	SHED	Poly-L-lactic acid (PLLA)	In-vivo (mice)	Transmission electron microscopy and immunohistochemistry	Odontoblast and endothelial-like cells can be differentiated from SHED
Demarco, 2010 [99]	DPSC	Poly-L-lactic acid (PLLA)	In-vivo (mice)	Immunohistochemistry	Differentiation was determined by evaluation of three putative odontoblastic markers (DSPP, DMP1, and MEPE)
Kodonas, 2012 [100]	DPSCs	- Type I atelocollagen honeycomb sponge (organic) - PLGA (synthetic)	In vivo (mini-pigs)	Histological and immunohistochemistry	The formation of new organic matrix deposits and odontoblast-like cell differentiation occurred.
Rosa, 2013 [83]	SHED	- Self-assembling peptide hydrogel - rhCollagen type I	In vivo (mice)	Histological and immunohistochemistry	Differentiation and proliferative activity to form microvessels and cellular density, expressed odontoblastic differentiation markers(DSPP, DMP-1, MEPE).
Wang Y, 2013 [101]	DPSC	Gelfoam	In vivo (dog)	Radiographic and histologic analyses	Generating pulp-like tissues containing dentin-like tissue and blood vessels.
Iohara K, 2014 [102]	DPSC	Atelocollagen	In vivo (dog)	Immunohistochemically evaluated	Regenerated pulp-like loose connective tissue with vasculature. Odontoblastlike cells attached to the dentinal wall, angiogenesis and re-innervation
Qu, 2014 [103]	DPSC	- NF-gelatin/MgP - NF-gelatin	In vitro In vivo (mice)	Immunohistochemical X-ray SEM ALP activity	NF-gelatin/MgP act better as scaffold than Nf-gelatin
Murakami, 2015 [104]	DPSCs/BMMSCs/ ADSCs	Atelocollagen	In-vivo (dog)	Immunohistochemistry	Neovascularization occurs, and nerve fibers form in the regenerated pulp tissue. The MDPSC transplantation showed a higher area of vascularization and innervation compared to the MBMSC and MADSC.
Y. S. Kwon, 2015 [105]	DPSC	Collagen hydrogel scaffold cross-linked with cinnamaldehyde (CA)	In vitro	Real-time polymerase chain reaction (PCR) gene expression analysis	Cross-linking of collagen scaffolds with CA is a new strategy for regenerative endodontic therapy regarding hDPC attachment, proliferation and differentiation.
Piva, 2017 [106]	DPSC	Medical-grade poly(L-lactide) (PLLA)	In vivo (mice)	Histology and Immunohistochemistry	Capable of differentiating into endothelial cells,
Widbiller, 2018 [107]	Extraction of dentin matrix protein (eDMP)	- Custom-made fibrin from fibrinogen and thrombin - Fibrin sealant - Self-assembling peptide (SAP) - Plasma rich in growth factor (PRGF)	In vivo (mice)	Histological and immunohistochemistry	eDMP + fibrin and fibrin sealant increased tissue formation than PRGF and SAP
Chang, 2020 [108]	DPSC	Autoclaved treated dentin matrix (a-TDM)	In vivo (mice and goats)	ALP activity spectrophotometer immunohistochemistry	a-TDM + DPSC effective in proliferating and differentiate
Chen H, 2020 [109]	DSC	Matrigel	In vivo (mice)	H&E staining	Microvessel formation, which resembled the natural pulp tissue.
Jang JH, 2020 [110]	DPSC	- Gelatin (GM)- based hemostatic hydrogels (GM) - Fibrin-based hemostatic hydrogels (FM)	In vivo (mini- pig)	Radiographic and histologic	- GM: absence of periapical inflammation and newly formed tertiary dentin with apex maturation - FM: exhibited higher incidences of inflammatory changes (periapical radiolucency and internal root resorption). - Showed microvasculature and odontoblastic layers

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
