# Peer review of "Tissue Engineering with Stem Cell from Human Exfoliated Deciduous Teeth (SHED) and Collagen Matrix, Regulated by Growth Factor in Regenerating the Dental Pulp"

_polymers, 2022, doi:10.3390/polym14183712_

Round 1
Reviewer 1 Report
I have reviewed the manuscript, I believe it does not fit within the scope of Polymers Journal. The review is very superficial, and does not present an adequate structure, The manuscript does not present enough novelty and will not be going to add any new information to the existing literature or any clinical significance.
Author Response
Response to Reviewer 1 Comments
Point 1: I have reviewed the manuscript, I believe it does not fit within the scope of Polymers Journal. The review is very superficial, and does not present an adequate structure, The manuscript does not present enough novelty and will not be going to add any new information to the existing literature or any clinical significance
Response 1: We thank the reviewers for their thorough reading of our manuscript and for providing constructive comments and suggestions. In accordance with the theme of the Special Issue from Polymers that was raised, namely "Multifunctional Collagen Based Biomaterials for Biomedical Applications", then in accordance with the field of science in dentistry, we introduced the theme of Regenerative Endodontics by utilizing tissue engineering technology. In this manuscript, we have tried to describe in detail the use of collagen biomaterials in tissue engineering technology combined with stem cells from human exfoliated primary teeth (SHED) regulated by growth factors, so that this technology can be used for the regeneration of dental pulp. Considering that research describing the use of SHED in pulp regeneration is still very limited and still needs to be developed.
Reviewer 2 Report
Title - TISSUE ENGINEERING WITH STEM CELL FROM HUMAN EXFOLIATED DECIDUOUS TEETH (SHED) AND COLLAGEN MATRIX, REGULATED BY GROWTH FACTOR IN REGENERATING THE DENTAL PULP
Journal – Polymers
Manuscript Number: polymers-1880583
Detailed Review Comments
Sugiaman et al. detail the most recent advances in TISSUE ENGINEERING for dental pulp regeneration employing human exfoliated deciduous tooth stem cells (SHED) and a collagen matrix, and their relationship to regulators such growth factors. The review is clear and succinct, covering all the bases that are necessary for a wide audience. I compared it to a chapter in a textbook since it has a thorough explanation of the subject and all the terminologies I needed to grasp. Some of the things I want to discuss are compiled below.
1) In general. Line 20 appears to be a mistake. Make sure to double-check and make any necessary changes.
2) The introduction is well-written and easy to understand.
3) Also, it would be helpful if the author included a table summarizing the most recent research in the topic. The titles of the columns might include information such as the year of publication, the type of dental stem cells employed, the treated condition, the success rate, and so on.
4) The review as a whole is well-structured and easy to read, making it perfect for readers with a specific focus. On the other side, although having excellent material, the review does not provide any visual summaries. The author has been urged to provide at least one figure and, if feasible, to distill all of the relevant data into a single graphic.
It is my view that the work as a whole would benefit from the aforementioned edits because of the rise in quality. In conclusion, the article is worthy of consideration for publishing in the journal Polymers.

Author Response
Point 1: In general. Line 20 appears to be a mistake. Make sure to double-check and make any necessary changes.
Response 1: We thank the reviewers for their thorough reading of our manuscript and for providing constructive comments and suggestions. Regarding the problem of typography, we have tried to revise what we have found throughout the manuscript. We apologize for the confusion.
Regarding the need to change the English we use in the manuscript, we will use mdpi editing services for proofreading. We hope this can improve the English used so that what we want to convey in this manuscript can be presented better and easier to understand.
Point 2: It would be helpful if the author included a table summarizing the most recent research in the topic. The titles of the columns might include information such as the year of publication, the type of dental stem cells employed, the treated condition, the success rate, and so on.
Response 2: Thank you for suggestion. We have supplemented this by adding a table summarizing the latest research in this topic on pages 7-9. With this table, it is hoped that it will become clearer.
Point 3: The review as a whole is well-structured and easy to read, making it perfect for readers with a specific focus. On the other side, although having excellent material, the review does not provide any visual summaries. The author has been urged to provide at least one figure and, if feasible, to distill all of the relevant data into a single graphic.
Response 3: Thank you for your valuable comment. I really appreciate it. We have added visual summaries on page 3 about “Tissue Engineering Technology in Dental Pulp Regeneration”.
Reviewer 3 Report
This is a good review on current concepts and applications of tissue engineering approach especially SHED 33 in endodontic treatment. I would recommend the author to add a summary sentence to clarify major topics of this review so that it is easier for readers to capture the information. This sentence can be "In this review, we focus on xxxx".
Many typographic errors need to be fixed before publication. For example, in the abstract: "A singl paragraph of about 200 words maximum" needs to be removed. The title of "Collagen Sacffold" needs to be aligned. Please pay attention to these error.
Author Response
Point 1: This is a good review on current concepts and applications of tissue engineering approach especially SHED 33 in endodontic treatment. I would recommend the author to add a summary sentence to clarify major topics of this review so that it is easier for readers to capture the information. This sentence can be "In this review, we focus on xxxx".
Response 1: We thank the reviewers for their thorough reading of our manuscript and for providing constructive comments and suggestions. We have added summary sentences at the end of the manuscript on page 10.
Point 2: Many typographic errors need to be fixed before publication. For example, in the abstract: "A singl paragraph of about 200 words maximum" needs to be removed. The title of "Collagen Sacffold" needs to be aligned. Please pay attention to these error.
Response 2: Thank you for suggestion. Regarding the problem of typography, we have tried to revise what we have found throughout the manuscript. We apologize for the confusion.
Regarding the need to change the English we use in the manuscript, we will use mdpi editing services for proofreading. We hope this can improve the English used so that what we want to convey in this manuscript can be presented better and easier to understand.
Round 2
Reviewer 1 Report
I am not convinced that this manuscript falls within the scope of Polymers Journal. The review only mentions a couple of journal statement about polymers. The review presents Stem cells from human exfoliated deciduous teeth in tissue engineering that has no relevance to polymers.